# CT features and histogram analysis of non-contrast images for differentiating malignant and benign mediastinal lymph nodes in Non-Small Cell Lung Cancer (NSCLC)

Pakorn Prakaikietikul[1], Yutthaphan Wannasopha[1]*, Juntima Euathrongchit[1‡], Apichat Tantraworasin[2,3‡]

1 Department of Radiology, Faculty of Medicine, Chiang Mai University, Chiang Mai, Thailand, 2 Clinical Epidemiology and Clinical Statistic Center, Faculty of Medicine, Chiang Mai University, Chiang Mai, Thailand, 3 Department of Surgery, Faculty of Medicine, Chiang Mai University, Chiang Mai, Thailand

☯ These authors contributed equally to this work.
‡ JE and AT also contributed equally to this work.
* wyutthaphan@gmail.com

## Abstract

### Objective

To evaluate the diagnostic value of CT features and histogram analysis in distinguishing between malignant and benign mediastinal lymph nodes in patients with non-small cell lung cancer (NSCLC).

### Method

This retrospective study analyzed non-contrast chest CT images from 40 NSCLC patients, comprising 80 pathology-proven mediastinal lymph nodes (46 benign, 34 metastasis). Morphologic features, including size, shape, margins, and internal composition, were independently assessed by two radiologists. Histogram analysis was conducted using the Synapse Vincent system with six parameters: mean attenuation, mean positive pixel (MPP), standard deviation (SD), skewness, kurtosis, and entropy. Statistical analysis included the Mann-Whitney test for continuous data, Fisher's exact test for categorical data, and receiver-operating characteristic (ROC) curve analysis to assess diagnostic accuracy, with statistical significance set at $p < 0.05$.

### Results

Malignant lymph nodes demonstrated significantly larger sizes ($p < 0.001$), ill-defined margins ($p = 0.024$), irregular shapes ($p < 0.001$), and the presence of necrotic areas ($p < 0.001$). A nodal size cutoff of 13.0 mm and volume of 1.632 ml were strongly associated with malignancy, yielding high diagnostic accuracy with sensitivities of 70.6% and 73.5% and specificities of 95.7% and 87.0%, respectively. Significant differences were observed between benign and malignant lymph nodes in several CT histogram parameters, including mean attenuation ($p = 0.004$), skewness ($p = 0.041$), kurtosis ($p = 0.005$), and entropy

provided the original author and source are credited.

**Data availability statement:** All relevant data are within the manuscript and its Supporting Information files.

**Funding:** The author(s) received no specific funding for this work.

**Competing interests:** The authors have declared that no competing interests exist.

(p < 0.001). The integrating all CT histogram parameters yielded an area under the curve (AUC) of 0.870 for differentiating between benign and malignant lymph nodes.

## Conclusion

The combination of morphologic CT features and CT histogram analysis offers a robust method for differentiating malignant from benign mediastinal lymph nodes in NSCLC patients, potentially enhancing diagnostic accuracy and informing treatment strategies.

## Introduction

Lung cancer is the second most common type of cancer worldwide, causing 1.8 million deaths annually [1]. The most frequently occurring cell type is a non-small cell lung cancer (NSCLC) [2]. Early detection through the use of low-dose CT scans and accurate lung cancer staging are important tools for appropriate treatment planning and improving patient survival [3,4]. Mediastinal staging is a crucial factor in designing treatment plans for NSCLC, and contralateral mediastinal lymph node involvement (N3) is a criterion for identification of an inoperable case [5]. Positron emission tomography integrated computed tomography (PET/CT) is a useful non-invasive mediastinal staging modality in NSCLC but it has a high false positive rate, which varies from 6.5% to 47.5% [4,6,7], especially in the regions with a high prevalence of inflammatory or infectious lung diseases such as pulmonary tuberculosis [3,4]. Additionally, PET/CT is not widely available in some countries, including Thailand. Due to the high false-positive rate of PET/CT, patients may undergo invasive procedures such as endobronchial ultrasound-guided fine needle aspiration (EBUS) and mediastinoscopy which carry a risk of complications more frequently [8]. Furthermore, transbronchial biopsy may not be able to reach all areas of mediastinal lymph nodes, as recommended by the International Association for the Study of Lung Cancer (IASLC) lymph node stations [9], such as prevascular, subaortic, paraaortic, paraesophageal, pulmonary ligament, and some of left paratracheal stations [10].

Recently, texture analysis, a key component of Radiomics, has been developed to assess and quantify tumor heterogeneity which cannot be detected by the human eye. Texture analysis can be applied to a variety of imaging modalities, including CT, Magnetic resonance imaging (MRI), and PET/CT. The first-order CT texture analysis (CTTA) involves creating a pixel intensity histogram in the area of interest, which provides quantitative measures of tissue heterogeneity. This approach is easily accessible and does not require expensive post-processing software. [11]. The concept behind CTTA is that the more heterogeneous a tumor is, the more biologically aggressive its behavior and the more resistance it will have to treatment [11].

The second-order and higher-order CTTA are more advanced techniques that provide more delicate details of the tumor, but they require specific software. A few studies have shown that the CTTA of mediastinal lymph nodes can differentiate between malignant and benign lymph nodes based on the hypothesis that metastatic lymph nodes are heterogeneous and of high density due to the loss of normal hilar fat. [10,12]. However, this is based on advanced CTTA techniques such as Run-length non-uniformity (RLNU), Gray-level non-uniformity (GLNU), and advanced metrics, while data about CT histogram is limited [7,13,14].

The CT histogram of mediastinal lymph nodes could be a significant method for mediastinal lymph nodal staging in NSCLC; however, there is limited data available, particularly in Thailand. This insufficiency motivated our search for an objective of this study, which was to evaluate the diagnostic value of the CT features and histogram analysis in differentiating between malignant and benign mediastinal lymph nodes in patients with NSCLC.

## Materials and methods

### Patient population

This study is a retrospective review of non-contrast chest CT scans of patients diagnosed with NSCLC. The population was searched using the keyword 'primary lung cancer' in the Envision program from January 2019 to June 2021, revealing 328 cases of primary lung cancer. Only 74 patients met inclusion criteria, which included the availability of pretreatment CT images in the CMU-PACS system, pathological confirmation of the lymph nodes through fine needle aspiration (FNA) or histology within an interval of less than three months and presence of ≥ 5 mm lymph nodes. Exclusion criteria included the absence of plain CT scans (30 patients) and poor image quality (4 patients), leaving 40 patients with 80 lymph nodes in this study (Fig 1). The study was approved by the institutional review board (**RAD-2565-09120**) and waived the requirement for informed consent due to its retrospective nature. The review of medical records was performed in accordance with institutional ethics review board guidelines.

### Data collection

The demographic data of patients, including gender, age, and pathologic reports, was collected from medical records. All protected health information (PHI) was securely stored in a password-protected computer in the Radiology Department. No traceable data that could identify the patients was collected.

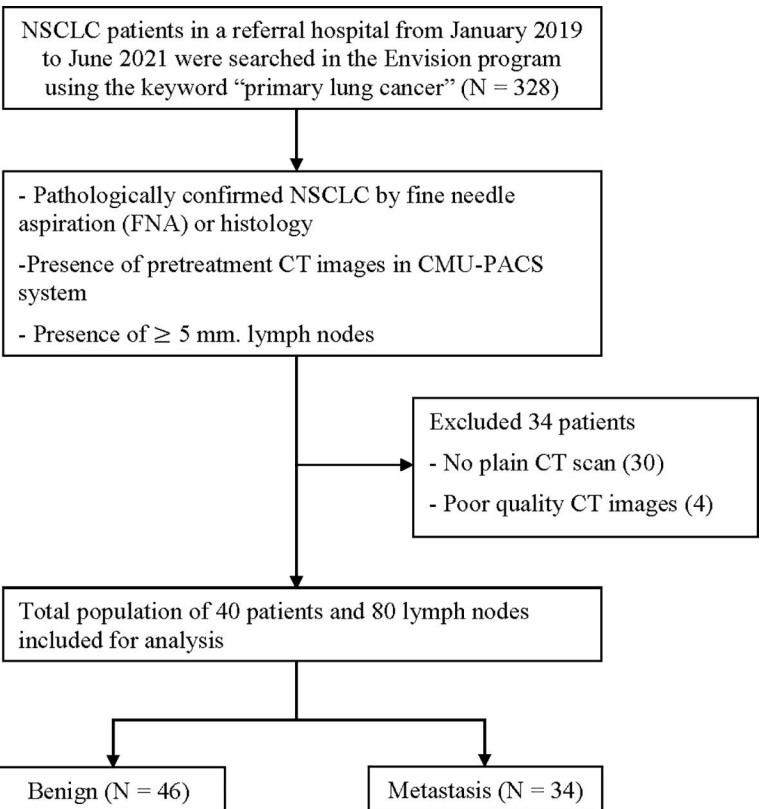

**Fig 1. Study workflow and patient selection.** NSCLC; non-small cell lung cancer, CMU-PACS; Chiang Mai University Picture archiving and communication system.

The non-contrast CT images were obtained from machines from three manufacturers, including SOMATOM Definition (Siemens Healthcare, Forchheim, Germany), SOMATOM Force (Siemens Healthcare, Forchheim, Germany), and Philips IQon (Philips Healthcare, Best, The Netherlands). CT studies were obtained on multidetector CT systems with a breath-held helical acquisition of the entire thorax, 120 kV and modified mAs. A collimation of 128 × 0.6 mm with z-flying focal spot technique was used. The images were reconstructed with 1-mm slice thickness and the field of view (FOV) that covered the entire thorax. Axial images at interested mediastinal lymph nodes were used for evaluation.

During surgery, the surgeons resected lymph nodes with some adipose connective tissue from the corresponding anatomic regions. The mediastinal lymph nodes were identified on the CT scan and correlated with the pathological reports by using size, location, and endo-bronchial ultrasound (EBUS) findings of the lymph nodes. The pathology of all the selected lymph nodes was verified by surgical pathology or fine needle aspiration from EBUS. The preoperative CT images of mediastinal lymph nodes were reviewed independently by two board-certified diagnostic radiologists; Y.W. and P.P. who had eleven years and five years of experience in chest imaging interpretation, respectively. Both were blinded to the pathological reports and clinical data. They focused on the following characteristics: size (short axis diameter), shape, margins, and internal composition, including the presence of calcification, necrosis, or fat components. Necrosis was defined as a focal area of low attenuation, appearing less dense than the surrounding tissues on CT scans [15].

The data from the CT scanner was sent to the workstation for CT histogram and the three-dimensional software automatically generated histograms of each node. The histogram features were computed by segmenting the whole volume of each lymph node using a three-dimensional volume of interest (VOI) approach. This method ensured comprehensive volumetric analysis, capturing spatial heterogeneity throughout the lymph node. The images were viewed using a standard mediastinal window setting with a level of 30 Hounsfield Units (HU) and a width of 400 HU (Fig 2).

A board-certified diagnostic radiologist (P.P.) segmented the mediastinal lymph nodes using a semi-automated method and analyzed them with six histogram measurements: mean attenuation, mean positive pixel (MPP), standard deviation (SD), skewness, kurtosis, and entropy using the Synapse Vincent system (Fujifilm Corp., Tokyo, Japan) (Fig 3). The histogram parameters used in this study are defined as follows [11]: Mean attenuation refers to the average density of the region of interest in Hounsfield units (HU); Mean positive pixel

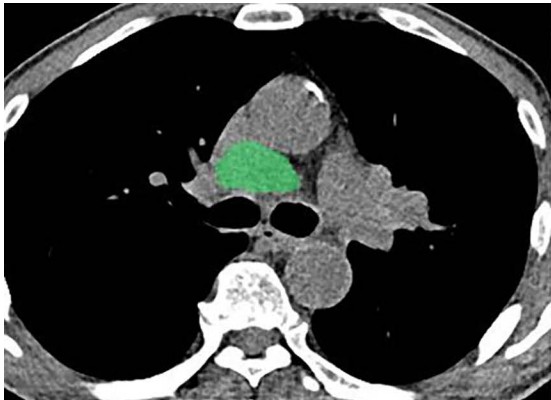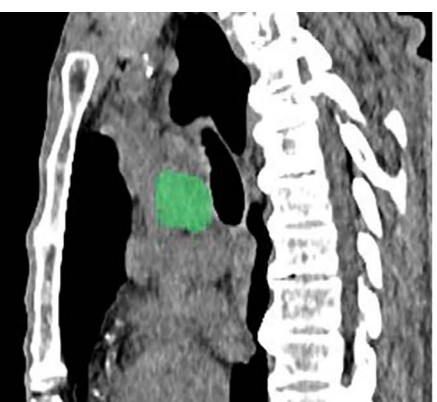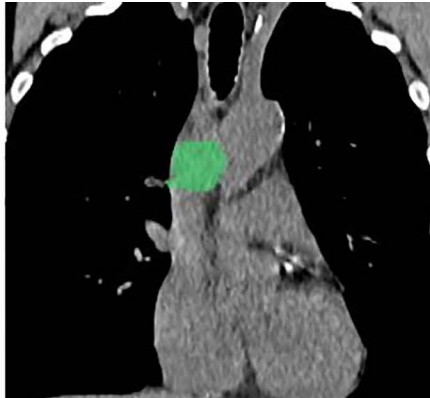

**Fig 2. The selected lymph node was labeled in green using semiautomated techniques and adjusted in axial, sagittal, and coronal views.**

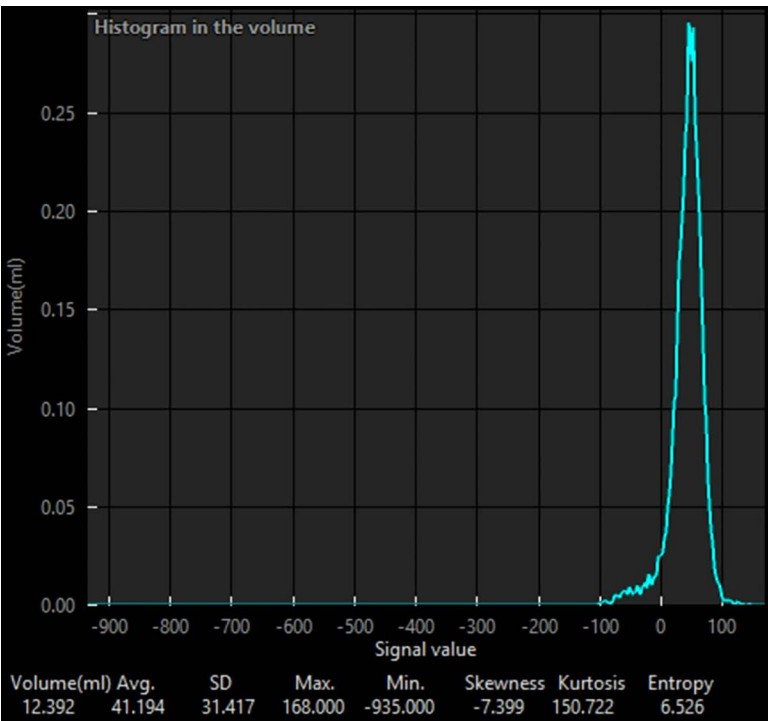

**Fig 3. The labeled area was analyzed using the Synapse 3D program, and the results were displayed as a histogram and in parameters.**

indicates the average density of the region of interest where the attenuation is greater than 0 HU; Skewness measures the asymmetry of the histogram, indicating whether the data distribution is skewed to the left or right; Kurtosis describes the peakedness or flatness of the histogram, reflecting the distribution of pixel intensities; and Entropy quantifies the irregularity or complexity of the histogram, with higher entropy values indicating greater heterogeneity.

## Method of data analysis

Descriptive analysis involved the calculation of frequencies and proportions for categorical variables. The interobserver agreement was reported using the Intraclass Correlation Coefficient (ICC) for continuous quantitative variables, and the kappa test was used for categorical variables. The ICC values below 0.5 suggest poor reliability, while values ranging from 0.5 to 0.75 indicate moderate reliability. Values between 0.75 and 0.9 indicate good reliability, and those exceeding 0.90 indicate excellent reliability [16]. The kappa values were evaluated based on the Viera classification, with the following interpretations: 0.01–0.20 indicating slight agreement, 0.21–0.40 indicating fair agreement, 0.41–0.60 indicating moderate agreement, 0.61–0.80 indicating substantial agreement, and 0.81–0.99 indicating almost perfect agreement [17]. The means with standard deviations (SDs) or medians with interquartile ranges (IQRs) of continuous variables were determined based on the distribution of the data. The collected data were evaluated using descriptive statistics, including absolute and relative frequencies. Group differences were analyzed using the Mann-Whitney test for continuous data and the Fisher's exact test for categorical data. Receiver-operating characteristic (ROC) curve analysis was used to assess diagnostic accuracy. Statistical significance was determined at a p-value of less than 0.05. The analyses were performed using IBM SPSS software version 25.

## Results

### General data

This study included 40 patients and 80 lymph nodes. The mean age of the patients was 66.9 ± 7.89 years, with a slightly higher proportion of male patients compared to female patients. Out of the 80 mediastinal lymph nodes, 46 were benign and 34 were metastatic nodes, as proven by pathology. Of the metastatic lymph nodes, 29 (36.3%) were adenocarcinoma and 5 (6.3%) were squamous cell carcinoma. The demographic data are demonstrated in Table 1.

### CT findings and histogram

The level of agreement between the two radiologists demonstrated excellent correlation in quantitative nodal size measurements (Intraclass Correlation Coefficient (ICC) = 0.94, p-value = 0.001). However, the agreement in qualitative nodal morphology evaluation showed fair to moderate agreement between the two radiologists (Kappa ranging from 0.412 to 0.756; p-value < 0.05) (Table 2).

Based on the analysis of the median values of nodal size and volume, it was found that the malignant lymph nodes were larger than the benign nodes. The benign lymph nodes had a median short axis of 8.0 mm (ranging from 6 to 10 mm), while the malignant lymph nodes had a median short axis of 14.0 mm (ranging from 12 to 20 mm). The difference between the two groups was statistically significant (p-value < 0.001). The median volume of the malignant group was 2.49 ml, which was significantly different from that of the benign group, which was 0.87 ml (p-value < 0.001).

When considering the morphologic appearances of the lymph nodes from the benign and malignant groups, analysis showed that the majority of the malignant nodes had an irregular shape (76.5%), ill-defined margins (73.5%), and contained necrotic components (50.0%), as compared to the benign nodes, with differences being statistically significant (p-value < 0.05). It was observed that the malignant lymph nodes did not have any internal fat component. However, the benign lymph nodes generally had an oval shape (65.2%), well-defined margins (93.5%), and a presence of internal hilar fat (17.4%). Table 3 shows the comparison between the CT findings from the malignant and benign mediastinal lymph nodes in patients with non-small cell lung cancer.

**Table 1. Demographic data and pathological reports of the investigated patient sample.**

| Demographic data N=40 | | n (percentage) |
|---|---|---|
| Age (years) (Mean ± SD) | | 66.9 ± 7.89 |
| Sex | Male | 23 (57.5) |
| | Female | 17 (42.5) |
| Pathological report | Squamous cell carcinoma | 5 (6.3) |
| | Adenocarcinoma | 29 (36.3) |
| | Benign lymph nodes | 46 (57.3) |

**Table 2. The interobserver agreement on nodal characteristics.**

| Nodal characteristics | Percent of agreement | Kappa | p-value |
|---|---|---|---|
| Shape | 80 | 0.600 | 0.019 |
| Margin | 80 | 0.412 | 0.050 |
| Component | 90 | 0.756 | 0.001 |

**Table 3. Comparison of nodal characteristics and CT histogram parameters for distinguishing between malignant and benign mediastinal lymph nodes in non-small cell lung cancer.**

| Parameters | Pathology | | p-value | AUC |
|---|---|---|---|---|
| | Benignity (n =46) | Malignancy (n = 34) | | (95% confidence interval) |
| **Size (mm)** | 8 (6–10) | 14 (12–20) | < 0.001 | 0.919 (0.862-0.976) |
| **Volume (ml)** | 0.87 (0.61–1.40) | 2.49 (1.56–9.29) | < 0.001 | 0.849 (0.759-0.939) |
| **Shapes** | | | < 0.001 | 0.725 (0.610-0.840) |
| Round | 2 (4.3) | 1 (2.9) | | |
| Oval | 30 (65.2) | 7 (20.6) | | |
| Irregular | 14 (30.4) | 26 (76.5) | | |
| **Margins** | | | 0.024 | 0.600 (0.471-0.728) |
| Well defined | 43 (93.5) | 25 (73.5) | | |
| Ill defined | 3 (6.5) | 9 (26.5) | | |
| **Components** | | | < 0.001 | 0.719 (0.596-0.842) |
| None | 32 (69.6) | 14 (41.2) | | |
| Fat | 8 (17.4) | 0 (0.0) | | |
| Calcification | 5 (10.9) | 3 (8.8) | | |
| Necrosis | 1 (2.2) | 17 (50.0) | | |
| **Histogram parameters** | | | | 0.870 (0.781-0.959) |
| Mean attenuation | 19.49 (8.96–28.20) | 29.17 (18.65–37.67) | 0.004 | |
| MPP | 41.88 ± 9.22 | 39.56 ± 8.09 | 1.758 | |
| SD | 50.82 (35.08–67.12) | 34.02 (24.97–71.66) | 0.083 | |
| Skewness | -2.88 (-4.26 - -0.90) | -4.54 (-6.16 - -0.99) | 0.041 | |
| Kurtosis | 16.11 (4.14–33.58) | 41.65 (8.76–81.44) | 0.005 | |
| Entropy | 7.02 ± 0.44 | 6.50 ± 0.49 | < 0.001 | |

*Data are presented as n (%), mean±SD or median (interquartile range)

A comparison of CT histogram parameters between benign and malignant nodal groups revealed significant differences in the mean attenuation, skewness, kurtosis, and entropy, with p-values of 0.004, 0.041, 0.005, and <0.001, respectively. However, there were no significant differences in the Maximum Pixel Percentage (MPP) and Standard Deviation (SD), with p-values of 1.758, and 0.083, respectively. Table 3 demonstrates the CT findings and histogram parameters, and the differences in histogram features between benign and malignant lymph nodes are illustrated using box plots (Fig 4). Additionally, the receiver-operating characteristics (ROC) curve analysis, incorporating all CT histogram parameters, yielded an area under the curve (AUC) of 0.870 for differentiating between benign and malignant lymph nodes (Fig 5).

The ROC curve analysis of qualitative features (shape, margin, and internal components) evaluated by radiologists for distinguishing benign from malignant lymph nodes showed an AUC of 0.827 (95% CI: 0.728–0.926). In comparison, quantitative histogram analysis achieved a higher AUC of 0.870 (95% CI: 0.781–0.959). However, the difference between the two methods was not statistically significant (p = 0.336).

The ROC curve analysis for the discrimination of benign versus malignant lymph nodes showed an AUC of 0.919 for nodal size and 0.849 for nodal volume (Fig 6). Further analysis revealed that a cut-off nodal size of 13.0 mm tended to indicate malignancy, with a sensitivity of 0.706, a specificity of 0.957, a Youden's Index of 0.662, an accuracy of 0.850,

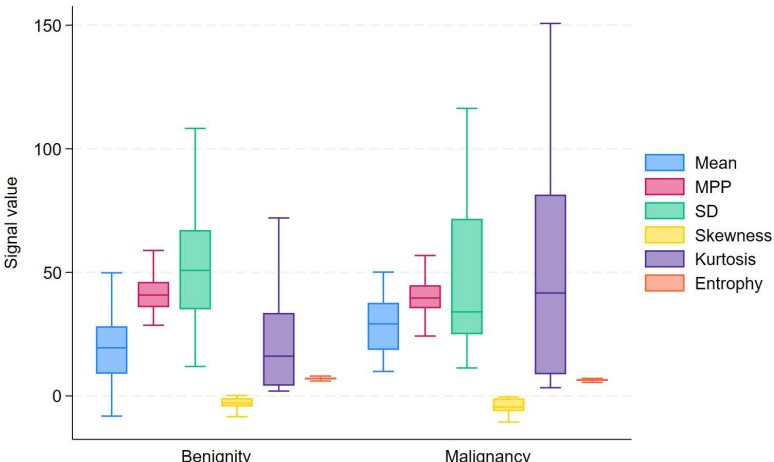

**Fig 4. Box plots analyzing the differences in histogram features between benign and malignant lymph nodes.**

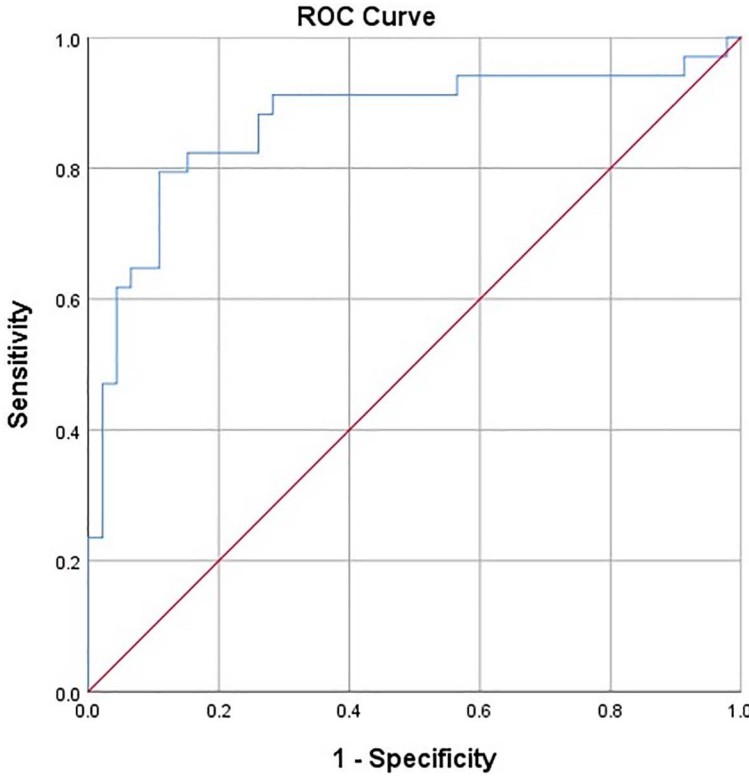

**Fig 5. Results of the receiver operating characteristic (ROC) curve analysis incorporating all CT histogram parameters for distinguishing between benign and malignant lymph nodes in NSCLC patients.**

and a positive likelihood ratio (LR+) of 16.23. Similarly, a cut-off nodal volume of 1.632 ml indicated a tendency toward malignancy, with a sensitivity of 0.735, a specificity of 0.870, a Youden's Index of 0.605, an accuracy of 0.813, and an LR+ of 5.64. The results are summarized in Table 4.

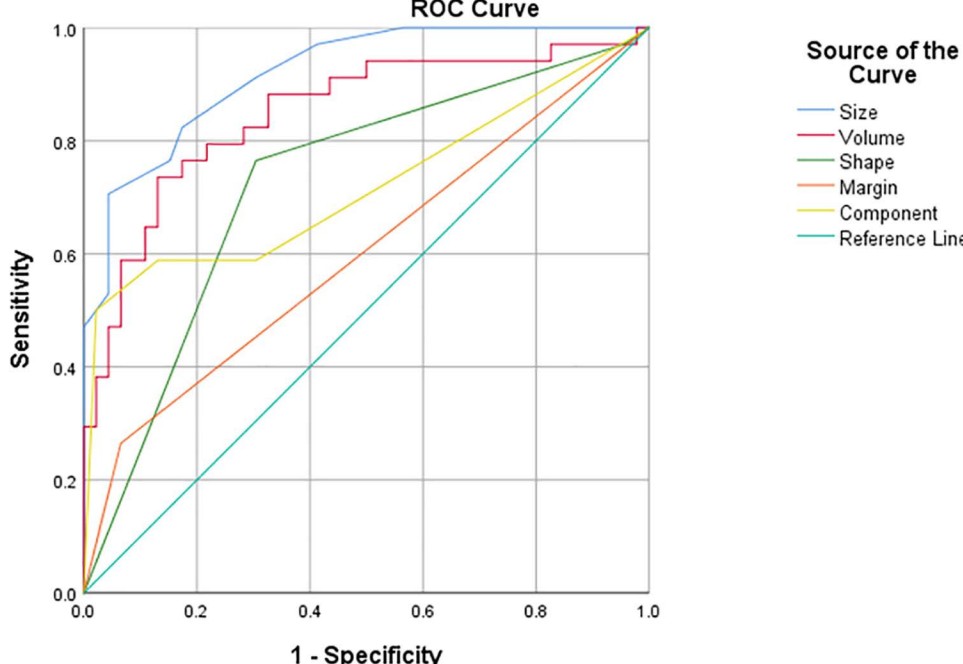

**Fig 6. Results of the receiver operating characteristic (ROC) curve analysis for distinguishing between benign and malignant lymph nodes in NSCLC patients.**

**Table 4. The cut-off values and diagnostic performance of nodal size and volume for differentiating benign and malignant lymph nodes.**

| Parameter | Cut-off | Sensitivity | Specificity | Youden's Index | Accuracy | LR+ |
|---|---|---|---|---|---|---|
| Short axis (mm) | 13.0 | 0.706 | 0.957 | 0.662 | 0.850 | 16.23 |
| Volume (ml) | 1.632 | 0.735 | 0.870 | 0.605 | 0.813 | 5.64 |

## Discussion

Our study showed that malignant lymph nodes were statistically significantly larger than benign ones. We used a cut-off nodal short axis of 13.0 mm and nodal volume of 1.632 ml to indicate malignant lymph nodes with precision, with an accuracy of 0.850 and 0.813, respectively. This supports most previously published studies that often use a short axis greater than 1 cm as the cutoff for a metastatic lymph node in NSCLC [18]. However, the sensitivity of using a short axis less than 1 cm to identify a normal lymph node is only between 46% and 57% [19,20], indicating that non-enlarged nodes can still be malignant. Meanwhile, emerging studies on normal lymph nodes suggest that relying solely on size to differentiate malignant from benign mediastinal nodes is not reliable, as these studies indicate a relatively high frequency of malignancy [21,22].

In our study, the majority of malignant nodes had an irregular shape (76.5%), ill-defined margins (73.5%), and contained necrotic components (50.0%). This could be due to the fact that benign nodes become more rounded or take on an irregular shape as a result of malignant infiltration. However, the contour of the node as seen on CT and MR imaging can have greater diagnostic value, as malignant nodes tend to show irregular or

ill-defined borders as a result of extracapsular disease extension or uneven growth [13,23]. On CT images, large metastatic nodes often display a heterogeneous appearance. Heterogeneity is a well-known characteristic of cancer, reflecting alterations in tissue architecture that are probably the result of various factors such as cell infiltration, abnormal angiogenesis, myxoid changes, and tumor necrosis [24,25]. A lower density nodal center seen on CT can be a result of necrosis. Even normal-sized nodes with necrosis should be considered as potentially malignant in these patients. Although low attenuation nodes can be a feature of malignant infiltration, they are also found in tuberculosis and fungal infections, which were not included in our study. Therefore, low attenuation nodes alone are not diagnostic of malignancy [13,23].

In our study, the presence of internal hilar fat was seen exclusively in benign lymph nodes. This is consistent with published studies which suggest that the presence of a fatty hilum is a commonly used parameter for differentiating between benign and malignant lymph nodes [26,27]. A typical normal lymph node has a uniform appearance, and the presence of hilar fat can be indicative of benignity, although this is not always the case [23].

Our study found that interobserver agreement for qualitative nodal morphology evaluation was fair to moderate (Kappa: 0.412–0.756; p < 0.05), suggesting limitations in subjective assessment of shape, margin, and internal components using visual inspection alone. Although the difference was not statistically significant (p = 0.336), quantitative histogram analysis achieved a higher AUC (0.870) than qualitative feature assessment by radiologists (AUC 0.827), indicating its potential for better distinguishing between benign and malignant lymph nodes. A possible explanation is that subjective image assessment relies on a radiologist's perception, making it prone to inconsistency and variability. In contrast, objective assessment uses quantitative metrics for a standardized approach [28]. Radiomics and quantitative imaging extract measurable data to characterize tissue properties. The histogram analysis provides additional quantitative insights into the heterogeneity of mediastinal lymph nodes in NSCLC patients, which are not achievable through conventional morphologic evaluation alone.

Some prior studies postulated that CT texture analysis is able to inform the histopathological tumor characteristics, and that the more heterogeneous a tumor is, the more biologically aggressive its behavior tends to be [11,29]. In this study, we utilized a first-order CT texture analysis and found significant differences in mean attenuation, skewness, kurtosis, and entropy between benign and malignant lymph nodes. In malignant lymph nodes, a higher density is expected due to the loss of fatty tissue in the hilar region, in contrast to benign lymph nodes. As expected, our results showed that malignant lymph nodes tend to have high mean attenuation and more negative skewness. However, the MPP in this study showed similarities between the two groups and was not statistically significant which may be attributed to several factors. One potential explanation is the heterogeneity within the malignant lymph nodes themselves. While malignant nodes are generally expected to have higher attenuation due to the loss of fatty tissue, the presence of necrotic areas, which were found in about 50% of the malignant nodes, could lower the overall attenuation [24], balancing out the mean values when compared to benign nodes. Additionally, benign lymph nodes can sometimes show increased attenuation due to reactive changes, such as inflammation or fibrosis, which can occur in response to infection or other non-malignant processes [27]. These changes can elevate the attenuation values of benign nodes, making the MPP values between benign and malignant nodes more comparable.

Several features associated with texture derived from CT and MRI images are able to reflect unique histopathological tumor characteristics at a microstructural level. These features

have proven useful in discriminating benign from malignant lymph nodes through the use of CT texture features [29]. The kurtosis of first-grade radiomics features reflects the peak apex degree of the gray distribution within the pixel. High kurtosis implies that the gray area is further away from the mean distribution, while low kurtosis indicates the opposite trend. Meanwhile, skewness reflects the offset characteristics and symmetry of the gray distribution relative to the mean. Higher skewness suggests higher tumor heterogeneity and a greater possibility of progression [30,31]. Presumably, higher heterogeneity in malignant lymph nodes is caused by the presence of tumor deposits, which results in increased CT heterogeneity that can be quantified using entropy-related second-order statistics [29].

According to hypotheses pertinent to heterogeneity in malignancy, it can be expected that malignant tumors would have a less peaked histogram, a low kurtosis value, and a high entropy value. However, our study found the opposite results. The CT histogram texture features were analyzed to identify factors associated with lymph node differentiation. The possible explanation is that the density of lymph nodes can vary significantly based on their anatomical location and the underlying pathology [6]. Benign nodes, especially in regions with a high prevalence of infectious agents, particularly in Thailand, might develop calcifications or other changes that increase their heterogeneity on CT images. On the other hand, malignant nodes often exhibit rapid growth and central necrosis, leading to more uniform low-density areas [24], which could explain the lower entropy observed in malignant nodes.

Another possible explanation for these results is the presence of reactive changes in benign lymph nodes, which can occur due to inflammatory processes or infections. These reactive changes can introduce variability in the texture of benign nodes [27], leading to higher kurtosis and lower entropy values. However, our study suggests that integrating all CT histogram parameters is valuable for differentiating between benign and malignant lymph nodes, with an AUC of 0.870. This finding supports the idea that evaluating all CT histogram parameters collectively, rather than individually, can enhance the accuracy of nodal evaluation. Additionally, the use of first-order CT texture analysis, which focuses on basic histogram features, may not capture the complex microstructural differences between malignant and benign nodes as effectively as higher-order texture analysis methods.

This study has some limitations that should be considered. Firstly, the sample size is small and the study is retrospective in nature, which may limit the generalizability of the findings. Additionally, during post-processing, imaging filters such as Laplacian or Gaussian bandpass filters were not applied, potentially affecting some histogram parameters. To address these limitations and enhance the robustness of the findings, further evaluation using second-order statistical CT texture analysis is recommended, as it may provide more detailed and comprehensive information.

## Conclusion

In summary, our study shows that morphologic CT features have the potential in distinguishing between malignant and benign mediastinal lymph nodes in patients with NSCLC. The histogram parameters obtained through CTTA can provide additional information that could assist in the differentiation between metastatic and benign lymph nodes. It is important to note that CTTA is not intended to replace tissue diagnosis, but it can be a valuable addition to the radiologist's confidence for the characterization of lymph nodes and can have a significant impact on patient management. Further studies are warranted to compare CTTA with other diagnostic modalities, such as EBUS/EUS and F18-FDG PET/CT, on a patient-by-patient basis using a larger sample size, in order to determine its full potential in clinical practice.

## Supporting information

**S1 table. The Cut-off Values and Diagnostic Performance of Histogram Parameters for Differentiating Benign and Malignant Lymph Nodes.**
(DOCX)

**S1 file. Raw data.**
(XLSX)

**S2 file. Ethics approval letter (RAD-2565–09120).**
(PDF)

## Acknowledgments

The authors would like to express our sincere gratitude to Mr. Yanakawee Khatsitalee for his assistance with statistical analysis and consultation for this study.

## Author contributions

**Conceptualization:** Pakorn Prakaikietikul, Yutthaphan Wannasopha, Juntima Euathrongchit, Apichat Tantraworasin.

**Data curation:** Pakorn Prakaikietikul, Yutthaphan Wannasopha.

**Formal analysis:** Pakorn Prakaikietikul, Yutthaphan Wannasopha, Apichat Tantraworasin.

**Funding acquisition:** Yutthaphan Wannasopha.

**Investigation:** Pakorn Prakaikietikul, Yutthaphan Wannasopha.

**Methodology:** Pakorn Prakaikietikul, Yutthaphan Wannasopha.

**Project administration:** Pakorn Prakaikietikul, Yutthaphan Wannasopha.

**Resources:** Pakorn Prakaikietikul, Yutthaphan Wannasopha.

**Software:** Pakorn Prakaikietikul, Yutthaphan Wannasopha.

**Supervision:** Yutthaphan Wannasopha, Juntima Euathrongchit.

**Validation:** Pakorn Prakaikietikul, Yutthaphan Wannasopha.

**Visualization:** Pakorn Prakaikietikul, Yutthaphan Wannasopha.

**Writing – original draft:** Pakorn Prakaikietikul, Yutthaphan Wannasopha.

**Writing – review & editing:** Pakorn Prakaikietikul, Yutthaphan Wannasopha, Juntima Euathrongchit, Apichat Tantraworasin.

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
