## [Decision Letter · Decision Letter 0]

8 Dec 2024

PONE-D-24-38941CT Features and Histogram Analysis for Differentiating Malignant and Benign Mediastinal Lymph Nodes in Non-Small Cell Lung Cancer (NSCLC)PLOS ONE

Dear Dr. Wannasopha,

Thank you for submitting your manuscript to PLOS ONE. After careful consideration, we feel that it has merit but does not fully meet PLOS ONE’s publication criteria as it currently stands. Therefore, we invite you to submit a revised version of the manuscript that addresses the points raised during the review process.

We look forward to receiving your revised manuscript.

Kind regards,

Lorenzo Faggioni, M.D., Ph.D.

Academic Editor

PLOS ONE

Reviewers' comments:

Reviewer's Responses to Questions

**Comments to the Author**

1. Is the manuscript technically sound, and do the data support the conclusions?

Reviewer #1: Yes

2. Has the statistical analysis been performed appropriately and rigorously?

Reviewer #1: Yes

3. Have the authors made all data underlying the findings in their manuscript fully available?

Reviewer #1: Yes

4. Is the manuscript presented in an intelligible fashion and written in standard English?

Reviewer #1: Yes

5. Review Comments to the Author

Reviewer #1: This manuscript explores an important area of radiology by attempting to use CT and histogram analysis to differentiate between malignant and benign lymph nodes in NSCLC. Below are specific comments and suggestions for improvement:

1. Size Measurement: Please clarify whether the lymph node size parameter refers to the long-axis or short-axis measurement.

2. In the Results section, the reporting of size statistics is inconsistent, with benign nodes described using median and malignant nodes using mean. Please standardize these metrics.

3. How was necrosis diagnosed in this study? As non-contrast CT is inherently limited in identifying necrosis, this methodology requires clarification and justification.

4. Was the evaluation performed by a single radiologist? To improve reliability, it is recommended that assessments be conducted by at least two radiologists, with an analysis of inter-observer agreement.

5. Please provide details on the statistical methods used for the multivariate analysis in the Materials and Methods section.

6. The multivariate analysis includes only histogram parameters. Would it not be more robust to include all parameters with significant differences in univariate analysis?

7. For the histogram analysis parameters, diagnostic cut-off values, sensitivity, specificity, and overall diagnostic accuracy should be provided to enhance the clinical applicability of your findings.

6. PLOS authors have the option to publish the peer review history of their article (what does this mean? ). If published, this will include your full peer review and any attached files.

**Do you want your identity to be public for this peer review?** For information about this choice, including consent withdrawal, please see our Privacy Policy .

Reviewer #1: No

---

## [Author Response · Author response to Decision Letter 1]

19 Jan 2025

Response to Reviewers

Dear Editors and Reviewers,

Thank you for allowing us to submit a revised version of our manuscript entitled “CT Features and Histogram Analysis for Differentiating Malignant and Benign Mediastinal Lymph Nodes in Non-Small Cell Lung Cancer (NSCLC)” for publication in “PLOS ONE”. We also thank the reviewers for their valuable comments that helped us improve our manuscript.

We have modified our manuscript incorporating the reviewers’ comments and we enclosed our response in a point-by-point manner.

We hope the revised manuscript will now be found suitable for publication in “PLOS ONE”. We look forward to hearing from you soon.

Reviewer #1: This manuscript explores an important area of radiology by attempting to use CT and histogram analysis to differentiate between malignant and benign lymph nodes in NSCLC. Below are specific comments and suggestions for improvement:

1. Size Measurement: Please clarify whether the lymph node size parameter refers to the long-axis or short-axis measurement.

Answer: Thank you for your insightful comment. In our study, the short-axis measurement was used to represent lymph node size. This clarification has been added to the manuscript in the Materials and Methods section on page 5, line 93-94.

2. In the Results section, the reporting of size statistics is inconsistent, with benign nodes described using median and malignant nodes using mean. Please standardize these metrics.

Answer: Thank you for your attention to this detail, and I apologize for the mistake. To ensure consistency, we have used the median to represent the average size of both benign and malignant nodes, as the data for both populations do not follow a normal distribution. This adjustment has been made in the Results section to maintain uniformity and accurately reflect the statistical characteristics of the data.

3. How was necrosis diagnosed in this study? As non-contrast CT is inherently limited in identifying necrosis, this methodology requires clarification and justification.

Answer: Thank you for your insightful comment regarding the diagnosis of necrosis. Necrotic regions within lymph nodes manifest as areas of low attenuation on non-contrast CT scans, appearing less dense than surrounding tissues. This characteristic reduction in density is suggestive of necrosis. In our study, we identified necrosis by observing these focal low-attenuation areas within lymph nodes on non-contrast CT images. This clarification has been added to the manuscript in the Materials and Methods section on page 5, line 95-96, along with the relevant code references.

4. Was the evaluation performed by a single radiologist? To improve reliability, it is recommended that assessments be conducted by at least two radiologists, with an analysis of inter-observer agreement.

Answer: Thank you for your valuable feedback. In our study, the preoperative CT images of mediastinal lymph nodes were independently and blindly reviewed by two board-certified diagnostic radiologists. The evaluation focused on the following characteristics: size, shape, margins, and internal composition.

The inter-observer agreement has been analyzed and added to the manuscript in the Results section on page 8, line 143-147, and is presented in Table 2.

5. Please provide details on the statistical methods used for the multivariate analysis in the Materials and Methods section.

6. The multivariate analysis includes only histogram parameters. Would it not be more robust to include all parameters with significant differences in univariate analysis?

Answer question number 5 and 6: Thank you for your valuable comments. We acknowledge the inaccuracies in our initial description of the statistical analysis. In diagnostic studies, multivariable analysis is typically employed to evaluate the collective predictive value of all variables or to identify key factors for inclusion in clinical prediction models.

However, in our study, the purpose of the analysis was not to develop a predictive model or assess all significant variables from the univariate analysis. Instead, we conducted Receiver-operating characteristic (ROC) curve analysis to calculate the area under the curve (AUC) for CT texture histogram parameters, focusing on their ability to distinguish malignant from benign mediastinal lymph nodes. This clarification has been incorporated into the Results section on page 9, lines 170-173 and is presented in Fig 3.

7. For the histogram analysis parameters, diagnostic cut-off values, sensitivity, specificity, and overall diagnostic accuracy should be provided to enhance the clinical applicability of your findings.

Answer: Thank you for your insightful feedback. Upon further analysis, we have revised the cut-off values for nodal size and volume. The cut-off for nodal size has been adjusted from 10.5 mm to 13.0 mm, and for nodal volume from 1.47 ml to 1.63 ml. These new cut-off values were selected because they are more strongly supported by Youden's Index and the positive likelihood ratio (LR+), making them the preferred indicators for malignancy. These have been added to the manuscript in the Results section on page 10, line 181-187, and is presented in Table 4.

For the remaining histogram parameters, our analysis revealed consistently low sensitivity, specificity, accuracy, Youden’s Index, and LR+, indicating limited utility in predicting malignancy. As a result, we have chosen to highlight only the cut-off values for size and volume in the main manuscript, while the full set of histogram parameter cut-offs is presented in the supplementary table.

We believe this approach enhances the clarity and clinical relevance of the manuscript. Please let us know if you would like further clarification or if additional revisions are necessary.

---

## [Decision Letter · Decision Letter 1]

21 Feb 2025

PONE-D-24-38941R1CT Features and Histogram Analysis for Differentiating Malignant and Benign Mediastinal Lymph Nodes in Non-Small Cell Lung Cancer (NSCLC)PLOS ONE

Dear Dr. Wannasopha,

Thank you for submitting your manuscript to PLOS ONE. While your revised manuscript contains several improvements over the original submission, we feel that some additional changes should be made before being considered for potential publication. Therefore, we invite you to submit a further revised version of the manuscript that addresses the points raised during the second review round.

We look forward to receiving your revised manuscript.

Kind regards,

Lorenzo Faggioni, M.D., Ph.D.

Academic Editor

PLOS ONE

Journal Requirements:

Reviewers' comments:

Reviewer's Responses to Questions

**Comments to the Author**

1. If the authors have adequately addressed your comments raised in a previous round of review and you feel that this manuscript is now acceptable for publication, you may indicate that here to bypass the “Comments to the Author” section, enter your conflict of interest statement in the “Confidential to Editor” section, and submit your "Accept" recommendation.

Reviewer #2: (No Response)

2. Is the manuscript technically sound, and do the data support the conclusions?

Reviewer #2: Partly

3. Has the statistical analysis been performed appropriately and rigorously?

Reviewer #2: Yes

4. Have the authors made all data underlying the findings in their manuscript fully available?

Reviewer #2: Yes

5. Is the manuscript presented in an intelligible fashion and written in standard English?

Reviewer #2: Yes

6. Review Comments to the Author

Reviewer #2: Dear Authors,

In this manuscript, the potential role of CT-derived features and histogram analysis in differentiating benign and malignant mediastinal lymph-nodes was investigated. The authors found that the combination of moprhologic fatures, such as nodal size, and histogram analysis, as mean attenuation, skeweness and kurtosis, yelded optimal performance in such clinical cenario. Please find below a list of specific comments to be addressed in order to improve the overall quality of the manuscript:

1. I would specify in the title that non-contrast CT images were analyzed.

2. Abstract: The methods heading of the abstract is extremely generic, please provide more information (e.g. who assessed the morphologic features? How was histogram analysis carried out?). More detail should be provided as well for statistical analysis.

3. Abstract: In the results, please provide specific p-values for each reported result.

4. Introduction: The introduction is overall well-written, however I wouldn't revise the radiomiocs/CTTA definition. This section is quite misleading, as it looks that histogram analysis is somehwat specific for CTTA, while it may be applied to all kind of images.

5.Inclusion and eclusion criteria should be clearly stated in the main text in the patient population subheading.

6. Please provide years of experience and field of expertise for the two diagnostic radiologists involved in the lymph nodes assessment.

7. Please clarify whether the whole volume of the lymph nodes was segmented (VOI) or only one slice (ROI) to compute the histogram features.

Who carrief out the semi-automated segmentation?

8. Please provide relevant references for histogram features definition

9. It is unclear whether whether all the mediastinal lymph nodes were segmented. If so, it is surprisingly low the number of segmented nodes. Please clarify.

10. Based on qualitative characteristics, I would have preferred if radiologists had been invited to differentiate between benign and malignant nodes. It would then be more interesting to compare the AUC of radiologists, using conventional characteristics, with that of the histogram analysis.

11. Please provide all specific p-values in the main text of the results.

12. Most of the analyzed morphological features are well-recognized characteristics of malignant nodes, it is not surprising that there is a difference in size or that only benign one have hilar fat. I would to keep the focuse on histogram analysis.

13. It would be interest to provide figure with box plot highlighting differences betweenm the two groups in histogram features.

7. PLOS authors have the option to publish the peer review history of their article (what does this mean? ). If published, this will include your full peer review and any attached files.

**Do you want your identity to be public for this peer review?** For information about this choice, including consent withdrawal, please see our Privacy Policy .

Reviewer #2: No

---

## [Author Response · Author response to Decision Letter 2]

12 Mar 2025

Dear Editors and Reviewers,

Thank you for allowing us to submit a revised version of our manuscript entitled “CT Features and Histogram Analysis of Non-Contrast Images for Differentiating Malignant and Benign Mediastinal Lymph Nodes in Non-Small Cell Lung Cancer (NSCLC)” for publication in “PLOS ONE”. We also thank the reviewers for their valuable comments that helped us improve our manuscript.

We have modified our manuscript incorporating the reviewers’ comments and we enclosed our response in a point-by-point manner.

We hope the revised manuscript will now be found suitable for publication in “PLOS ONE”. We look forward to hearing from you soon.

Reviewer #2: In this manuscript, the potential role of CT-derived features and histogram analysis in differentiating benign and malignant mediastinal lymph-nodes was investigated. The authors found that the combination of morphologic features, such as nodal size, and histogram analysis, as mean attenuation, skeweness and kurtosis, yielded optimal performance in such clinical scenario. Please find below a list of specific comments to be addressed in order to improve the overall quality of the manuscript:

1. I would specify in the title that non-contrast CT images were analyzed.

Answer: Thank you for your insightful suggestion. To clarify the methodology and enhance the specificity of the title, we have revised it as follows:

Revised Title: CT Features and Histogram Analysis of Non-Contrast Images for Differentiating Malignant and Benign Mediastinal Lymph Nodes in Non-Small Cell Lung Cancer (NSCLC)

This modification explicitly indicates the use of non-contrast CT images, ensuring that readers are fully aware of the imaging modality employed in this study. We believe this change improves the precision and relevance of the title.

This modification has been added to the manuscript in the title on page 1, line 1-2.

2. Abstract: The methods heading of the abstract is extremely generic, please provide more information (e.g. who assessed the morphologic features? How was histogram analysis carried out?). More detail should be provided as well for statistical analysis.

Answer: Thank you for your valuable feedback. We agree that the methods section of the abstract would benefit from additional specificity to provide a clearer understanding of our study's approach. We have revised the methods section as follows:

Method: This retrospective study analyzed non-contrast chest CT images from 40 NSCLC patients, comprising 80 pathology-proven mediastinal lymph nodes (46 benign, 34 metastasis). Morphologic features, including size, shape, margins, and internal composition, were independently assessed by two radiologists. Histogram analysis was conducted using the Synapse Vincent system with six parameters: mean attenuation, mean positive pixel (MPP), standard deviation (SD), skewness, kurtosis, and entropy. Statistical analysis included the Mann-Whitney test for continuous data, Fisher’s exact test for categorical data, and receiver-operating characteristic (ROC) curve analysis to assess diagnostic accuracy, with statistical significance set at p < 0.05.

3. Abstract: In the results, please provide specific p-values for each reported result.

Answer: Thank you for your helpful suggestion. To enhance the clarity and scientific rigor of the results section, we have added specific p-values for each reported finding in the abstract on page 2, line 14-15.

4. Introduction: The introduction is overall well-written; however, I wouldn't revise the radiomics/CTTA definition. This section is quite misleading, as it looks that histogram analysis is somewhat specific for CTTA, while it may be applied to all kind of images.

Answer: Thank you for pointing out this important distinction. We appreciate your insight into the broader applicability of histogram analysis beyond CT texture analysis (CTTA). We have revised the relevant section of the introduction to clarify this point on page 4, lines 48-56

5. Inclusion and exclusion criteria should be clearly stated in the main text in the patient population subheading.

Answer: Thank you for your valuable feedback. We agree that clearly stating the inclusion and exclusion criteria would enhance the transparency and reproducibility of our study. We have revised the "Patient Population" section under the "Materials and Methods" heading to explicitly include these criteria on page 5, lines 75-81 and added patient flow chart (Fig 1).

6. Please provide years of experience and field of expertise for the two diagnostic radiologists involved in the lymph nodes assessment

Answer: Thank you for this insightful suggestion. We agree that specifying the years of experience and fields of expertise of the radiologists enhances the credibility and reliability of the image assessment process. We have revised the "Materials and Methods" section accordingly. The revised text is added to the manuscript in the Materials and Methods section on page 6, line 107-110.

7. Please clarify whether the whole volume of the lymph nodes was segmented (VOI) or only one slice (ROI) to compute the histogram features. Who carried out the semi-automated segmentation?

Answer: Thank you for seeking clarification on the segmentation approach. We appreciate the opportunity to provide additional details. We have revised the "Materials and Methods" section to specify the segmentation approach and the semi-automated segmentation process using the Synapse Vincent system (Fujifilm Corp., Tokyo, Japan) on page 6-7, lines 115-118 and on page 7, lines 121-124.

8. Please provide relevant references for histogram features definition

Answer: Thank you for your insightful suggestion. We agree that providing relevant references for the definitions of histogram features will enhance the scientific rigor and credibility of the study. We have revised the "Materials and Methods" section to include the study by Lubner MG et al. [11] as a reference for the histogram features used in this study on page 7, line 125.

9. It is unclear whether all the mediastinal lymph nodes were segmented. If so, it is surprisingly low the number of segmented nodes. Please clarify.

Answer: Thank you for your observation and for giving us the opportunity to clarify this point. We agree that providing a clear explanation of the segmentation process and the number of lymph nodes analyzed will enhance the transparency of our study.

Lymph node selection and segmentation:

All mediastinal lymph nodes from the enrolled patients were evaluated, but only those that met the following criteria were included in the analysis:

Lymph nodes with a short-axis diameter of ≥ 5 mm on non-contrast chest CT images.

Lymph nodes that were clearly visible and could be accurately segmented using semi-automated techniques.

Pathologically confirmed as either benign or malignant through fine needle aspiration (FNA) or histology within an interval of less than three months.

Reason for low number of segmented nodes:

Our hospital is a major tertiary referral center in northern Thailand, treating a high volume of advanced-stage lung cancer cases. Most of these cases are inoperable, preventing access to pathological confirmation of mediastinal lymph nodes.

For cases where surgical pathology or fine needle aspiration from EBUS was available, we ensured that the identified lymph nodes corresponded precisely to those seen on preoperative CT based on size and location. Any uncertain cases were excluded to maintain the robustness of our results. Additionally, some operable cases had preoperative CT scans from referring hospitals with protocols incompatible with our CT system, preventing histogram analysis using the Synapse Vincent system.

Therefore, the relatively low number of segmented lymph nodes resulted from our strict inclusion criteria, which required both radiologic visibility and pathological confirmation. This approach was necessary to ensure high diagnostic accuracy and reliable statistical analysis.

Out of the initial pool of lymph nodes, 80 nodes (46 benign and 34 malignant) met the inclusion criteria and were segmented using a three-dimensional volume of interest (VOI) approach. Nodes that were too small (< 5 mm), poorly visible, or lacked pathological confirmation were excluded from the analysis to maintain segmentation accuracy and pathological correlation.

10. Based on qualitative characteristics, I would have preferred if radiologists had been invited to differentiate between benign and malignant nodes. It would then be more interesting to compare the AUC of radiologists, using conventional characteristics, with that of the histogram analysis.

Answer: Thank you for your insightful suggestion. We agree that comparing the diagnostic performance of radiologists using conventional qualitative characteristics with that of histogram analysis would provide valuable insights. This comparison has been analyzed and added to the manuscript in the Results section on page 11, lines 204-208.

11. Please provide all specific p-values in the main text of the results.

Answer: Thank you for your constructive feedback. We appreciate your attention to detail, as reporting specific p-values enhances the transparency and statistical rigor of our results. We have revised the "Results" section to include all relevant p-values for each reported finding.

12. Most of the analyzed morphological features are well-recognized characteristics of malignant nodes, it is not surprising that there is a difference in size or that only benign one have hilar fat. I would to keep the focus on histogram analysis.

Answer: Thank you for your valuable insight. We agree that the novelty of this study lies in the utilization of histogram analysis, and maintaining the focus on these advanced features will enhance the scientific impact. To address your comment, we have revised the discussion section to emphasize the significance of histogram analysis. The revised text has been added to the manuscript in the discussion section on pages 13-16, lines 252-313.

13. It would be interest to provide figure with box plot highlighting differences between the two groups in histogram features.

Answer: Thank you for your valuable recommendation. We have added box plots in Figure 4 of the results section to illustrate the differences in histogram features between benign and malignant lymph nodes. We trust that this will make it easier for readers to understand.

We believe this approach enhances the clarity and clinical relevance of the manuscript. Please let us know if you would like further clarification or if additional revisions are necessary.

---

## [Decision Letter · Decision Letter 2]

14 Mar 2025

CT Features and Histogram Analysis of Non-Contrast Images for Differentiating Malignant and Benign Mediastinal Lymph Nodes in Non-Small Cell Lung Cancer (NSCLC)

PONE-D-24-38941R2

Dear Dr. Wannasopha,

We’re pleased to inform you that your manuscript has been judged scientifically suitable for publication and will be formally accepted for publication once it meets all outstanding technical requirements.

Kind regards,

Lorenzo Faggioni, M.D., Ph.D.

Academic Editor

PLOS ONE

Reviewers' comments:

Reviewer's Responses to Questions

**Comments to the Author**

1. If the authors have adequately addressed your comments raised in a previous round of review and you feel that this manuscript is now acceptable for publication, you may indicate that here to bypass the “Comments to the Author” section, enter your conflict of interest statement in the “Confidential to Editor” section, and submit your "Accept" recommendation.

Reviewer #2: All comments have been addressed

2. Is the manuscript technically sound, and do the data support the conclusions?

Reviewer #2: (No Response)

3. Has the statistical analysis been performed appropriately and rigorously?

Reviewer #2: Yes

4. Have the authors made all data underlying the findings in their manuscript fully available?

Reviewer #2: Yes

5. Is the manuscript presented in an intelligible fashion and written in standard English?

Reviewer #2: Yes

6. Review Comments to the Author

Reviewer #2: Dear Authors, thanks for having so kindly addressed all of my comments and congratulations for your work, of great scientific value.

7. PLOS authors have the option to publish the peer review history of their article (what does this mean? ). If published, this will include your full peer review and any attached files.

**Do you want your identity to be public for this peer review?** For information about this choice, including consent withdrawal, please see our Privacy Policy .

Reviewer #2: No

---

## [Editor Report · Acceptance letter]

PONE-D-24-38941R2

PLOS ONE

Dear Dr. Wannasopha,

I'm pleased to inform you that your manuscript has been deemed suitable for publication in PLOS ONE. Congratulations! Your manuscript is now being handed over to our production team.

Kind regards,

on behalf of

Dr. Lorenzo Faggioni

Academic Editor

PLOS ONE